# Tracking cell lineages in 3D by incremental deep learning

**Ko Sugawara[1,2]\*, Çağrı Çevrim[1,2], Michalis Averof[1,2]\***

[1]Institut de Génomique Fonctionnelle de Lyon (IGFL), École Normale Supérieure de Lyon, Lyon, France; [2]Centre National de la Recherche Scientifique (CNRS), Paris, France

**Abstract** Deep learning is emerging as a powerful approach for bioimage analysis. Its use in cell tracking is limited by the scarcity of annotated data for the training of deep-learning models. Moreover, annotation, training, prediction, and proofreading currently lack a unified user interface. We present ELEPHANT, an interactive platform for 3D cell tracking that addresses these challenges by taking an incremental approach to deep learning. ELEPHANT provides an interface that seamlessly integrates cell track annotation, deep learning, prediction, and proofreading. This enables users to implement cycles of incremental learning starting from a few annotated nuclei. Successive prediction-validation cycles enrich the training data, leading to rapid improvements in tracking performance. We test the software's performance against state-of-the-art methods and track lineages spanning the entire course of leg regeneration in a crustacean over 1 week (504 timepoints). ELEPHANT yields accurate, fully-validated cell lineages with a modest investment in time and effort.

## Introduction

Recent progress in deep learning has led to significant advances in bioimage analysis (*Moen et al., 2019*; *Ouyang et al., 2018*; *Weigert et al., 2018*). As deep learning is data-driven, it is adaptable to a variety of datasets once an appropriate model architecture is selected and trained with adequate data (*Moen et al., 2019*). In spite of its powerful performance, deep learning remains challenging for non-experts to utilize, for three reasons. First, pre-trained models can be inadequate for new tasks and the preparation of new training data is laborious. Because the quality and quantity of the training data are crucial for the performance of deep learning, users must invest significant time and effort in annotation at the start of the project (*Moen et al., 2019*). Second, an interactive user interface for deep learning, especially in the context of cell tracking, is lacking (*Kok et al., 2020*; *Wen et al., 2021*). Third, deep learning applications are often limited by accessibility to computing power (high-end GPU).

We have addressed these challenges by establishing ELEPHANT (Efficient learning using sparse human annotations for nuclear tracking), an interactive web-friendly platform for cell tracking, which seamlessly integrates manual annotation with deep learning and proofreading of the results. ELEPHANT implements two algorithms optimized for incremental deep learning using sparse annotations, one for detecting nuclei in 3D and a second for linking these nuclei across timepoints in 4D image datasets. Incremental learning allows models to be trained in a stepwise fashion on a given dataset, starting from sparse annotations that are incrementally enriched by human proofreading, leading to a rapid increase in performance (*Figure 1*). ELEPHANT is implemented as an extension of Mastodon (https://github.com/mastodon-sc/mastodon; *Mastodon Science, 2021*), an open-source framework for large-scale tracking deployed in Fiji (*Schindelin et al., 2012*). It implements a client-server architecture, in which the server provides a deep learning environment equipped with sufficient GPU (*Figure 1—figure supplement 1*).

**\*For correspondence:**
ko.sugawara@ens-lyon.fr (KS);
michalis.averof@ens-lyon.fr (MA)

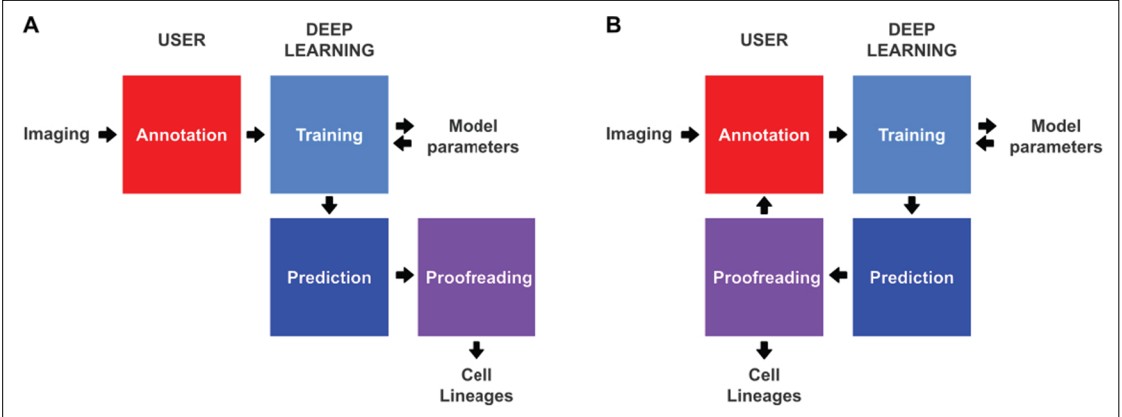

**Figure 1.** Conventional and incremental deep learning workflows for cell tracking. (**A**) Schematic illustration of a typical deep learning workflow, starting with the annotation of imaging data to generate training datasets, training of deep learning models, prediction by deep learning and proofreading. (**B**) Schematic illustration of incremental learning with ELEPHANT. Imaging data are fed into a cycle of annotation, training, prediction, and proofreading to generate cell lineages. At each iteration, model parameters are updated and saved. This workflow applies to both detection and linking phases (see *Figures 2A and 4A*).

The online version of this article includes the following figure supplement(s) for figure 1:

**Figure supplement 1.** ELEPHANT client-server architecture.

**Figure supplement 2.** Block diagram of ELEPHANT tracking workflow.

## Results and discussion

ELEPHANT employs the tracking-by-detection paradigm (*Maška et al., 2014*), which involves initially the *detection* of nuclei in 3D and subsequently their *linking* over successive timepoints to generate tracks. In both steps, the nuclei are represented as ellipsoids, using the data model of Mastodon (*Figure 2A* and Figure 4A). We use ellipsoids for annotation because ellipsoids allow rapid and efficient training and prediction, compared with more complex shapes. This is essential for interactive

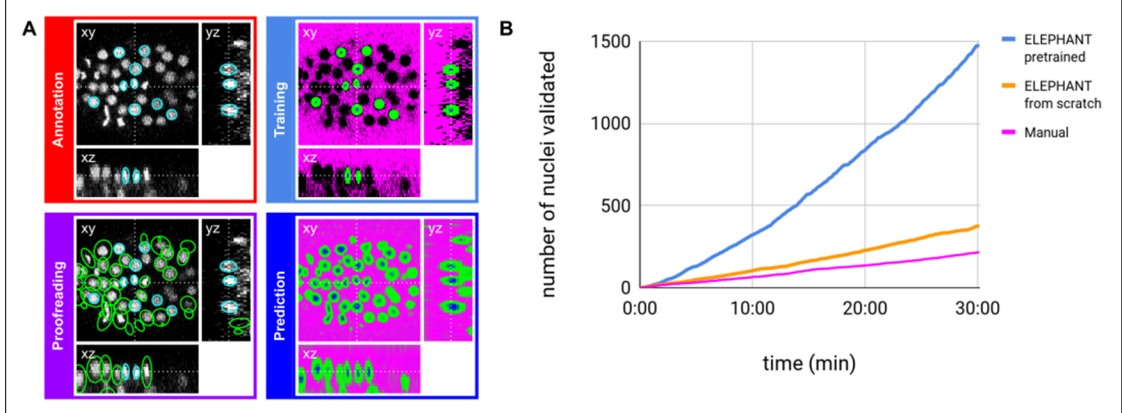

**Figure 2.** ELEPHANT detection workflow. (**A**) Detection workflow, illustrated with orthogonal views on the CE1 dataset. Top left: The user annotates nuclei with ellipsoids in 3D; newly generated annotations are colored in cyan. Top right: The detection model is trained with the labels generated from the sparse annotations of nuclei and from the annotation of *background* (in this case by intensity thresholding); *background*, *nucleus center*, *nucleus periphery* and unlabelled voxels are indicated in magenta, blue, green, and black, respectively. Bottom right: The trained model generates voxel-wise probability maps for *background* (magenta), *nucleus center* (blue), or *nucleus periphery* (green). Bottom left: The user validates or rejects the predictions; predicted nuclei are shown in green, predicted and validated nuclei in cyan. (**B**) Comparison of the speed of detection and validation of nuclei on successive timepoints in the CE1 dataset, by manual annotation (magenta), semi-automated detection without a pre-trained model (orange) and semi-automated detection using a pre-trained model (blue) using ELEPHANT.

The online version of this article includes the following figure supplement(s) for figure 2:

**Figure supplement 1.** 3D U-Net architecture for detection.

**Figure supplement 2.** Proofreading in detection.

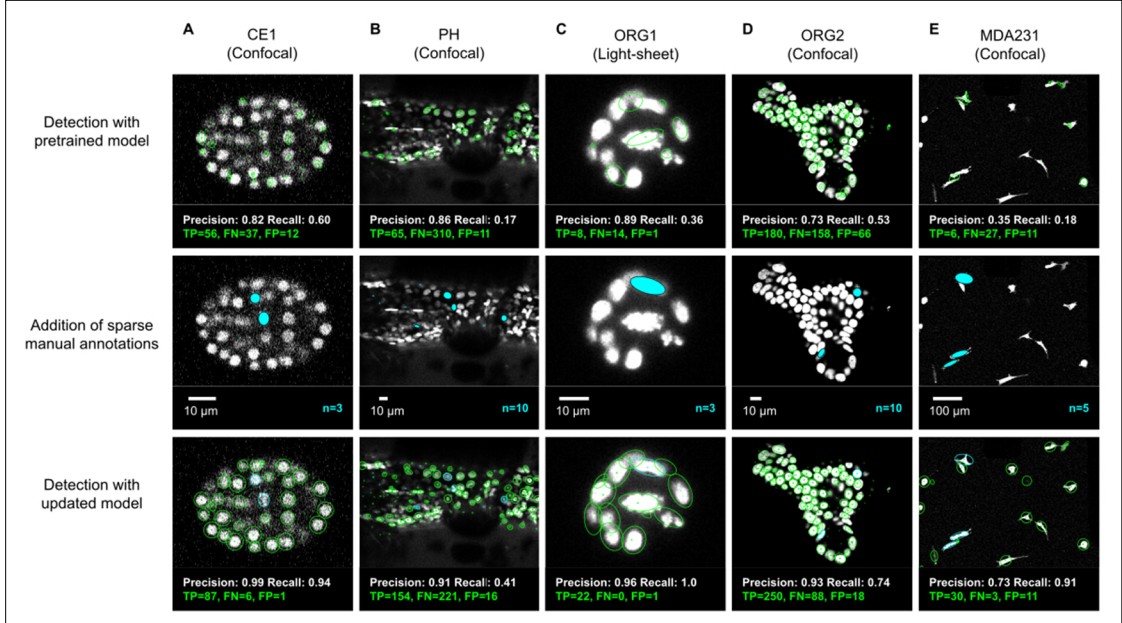

**Figure 3.** ELEPHANT detection with sparse annotations. Detection results obtained using ELEPHANT with sparse annotations on five image datasets recording the embryonic development of *C. elegans* (CE1 dataset, **A**), leg regeneration in the crustacean *P. hawaiensis* (PH dataset, **B**), human intestinal organoids (ORG1 and ORG2, **C and D**), and human breast carcinoma cells (MDA231, **E**). CE1, PH, ORG2, and MDA231 were captured by confocal microscopy; ORG1 was captured by light sheet microscopy. Top: Detection results using models that were pre-trained on diverse annotated datasets, excluding the test dataset (see *Supplementary file 3*). Precision and Recall scores are shown at the bottom of each panel, with the number of true positive (TP), false positive (FP), and false negative (FN) predicted nuclei. Middle: Addition of sparse manual annotations for each dataset. n: number of sparse annotations. Scale bars: 10 μm. Bottom: Detection results with an updated model that used the sparse annotations to update the pre-trained model. Precision, Recall, TP, FP, and FN values are shown as in the top panels.

The online version of this article includes the following figure supplement(s) for figure 3:

**Figure supplement 1.** Comparing detection predictions of ELEPHANT and StarDist3D.

**Figure supplement 2.** Evaluation of overfitting in detection using ELEPHANT.

deep learning. In the detection phase, voxels are labelled as *background*, *nucleus center* or *nucleus periphery*, or left unlabelled (*Figure 2A*, top right). The *nucleus center* and *nucleus periphery* labels are generated by the annotation of nuclei, and the *background* can be annotated either manually or by intensity thresholding. Sparse annotations (e.g. of a few nuclei in a single timepoint) are sufficient to start training. A U-Net convolutional neural network (U-Net CNN; *Cicek et al., 2016*; *Ronneberger et al., 2015*, *Figure 2—figure supplement 1*) is then trained on these labels (ignoring the unlabelled voxels) to generate voxel-wise probability maps for *background*, *nucleus center*, or *nucleus periphery*, across the entire image dataset (*Figure 2A*, bottom right). Post-processing on these probability maps yields predictions of nuclei which are available for visual inspection and proofreading (validation or rejection of each predicted nucleus) by the user (*Figure 2A*, bottom left). Human-computer interaction is facilitated by color coding of the annotated nuclei as predicted (green), accepted (cyan), or rejected (magenta) (see *Figure 2—figure supplement 2*), based on the proofreading. The cycles of training and prediction are rapid because only a small amount of training data are added each time (in the order of seconds, see *Supplementary file 1*). As a result, users can enrich the annotations by proofreading the output almost simultaneously, enabling incremental training of the model in an efficient manner.

We evaluated the detection performance of ELEPHANT on diverse image datasets capturing the embryonic development of *Caenorhabditis elegans* (CE1), leg regeneration in the crustacean *Parhyale hawaiensis* (PH), human intestinal organoids (ORG1 and ORG2) and human breast carcinoma cells (MDA231) by confocal or light sheet microscopy (*Figure 3A–E*). First, we tested the performance of a generic model that had been pre-trained with various annotated image datasets (*Figure 3A–E* top). We then annotated 3–10 additional nuclei or cells on each test dataset (*Figure 3A–E* middle) and re-trained the model. This resulted in greatly improved detection performance (*Figure 3A–E*),

**Table 1.** Performance of ELEPHANT on the Cell Tracking Challenge dataset.

Performance of ELEPHANT compared with two state-of-the-art algorithms, using the metrics of the Cell Tracking Challenge on unseen CE datasets. ELEPHANT outperforms the other methods in detection and linking accuracy (DET and TRA metrics); it performs less well in segmentation accuracy (SEG).

|  | ELEPHANT | KTH-SE | KIT-Sch-GE |
|---|---|---|---|
|  | (IGFL-FR) |  |  |
| SEG | 0.631 | 0.662 | 0.729 |
| TRA | 0.975 | 0.945 | 0.886 |
| DET | 0.979 | 0.959 | 0.930 |

showing that a very modest amount of additional training on a given dataset can yield rapid improvements in performance. We find that sparsely trained ELEPHANT detection models have a comparable performance to state-of-the-art software (*Figure 3—figure supplement 1*) and fully trained ELEPHANT models outperform most tracking software (*Table 1*).

We also investigated whether training with sparse annotations could cause overfitting of the data in the detection model, by training the detection model using sparse annotations in dataset CE1 and calculating the loss values using a second, unseen but similar dataset (see Materials and methods). The training and validation learning curves did not show any signs of overfitting even after a large amount of training (*Figure 3—figure supplement 2*). The trained model could detect nuclei with high precision and recall both on partially seen data (CE1) and unseen data (CE2). A detection model that has been pre-trained with diverse image datasets is available to users as a starting point for tracking on new image data (see Materials and methods).

In the linking phase, we found that nearest neighbor approaches for tracking nuclei over time (*Crocker and Grier, 1996*) perform poorly in challenging datasets when the cells are dividing; hence we turned to optical flow modeling to improve linking (*Amat et al., 2013*; *Horn and Schunck, 1981*; *Lucas and Kanade, 1981*). A second U-Net CNN, optimized for optical flow estimation (*Figure 4—figure supplement 1*), is trained on manually generated/validated links between nuclei in successive timepoints (*Figure 4A*, top left). Unlabelled voxels are ignored, hence training can be performed on sparse linking annotations. The flow model is used to generate voxel-wise 3D flow maps, representing predicted x, y and z displacements over time (*Figure 4A*, bottom right), which are then combined with nearest neighbor linking to predict links between the detected nuclei (see Materials and methods). Users proofread the linking results to finalize the tracks and to update the labels for the next iteration of training (*Figure 4A*, bottom left).

We evaluated the linking performance of ELEPHANT using two types of 4D confocal microscopy datasets in which nuclei were visualized by fluorescent markers: the first type of dataset captures the embryonic development of *Caenorhabditis elegans* (CE datasets), which has been used in previous studies to benchmark tracking methods (*Murray et al., 2008*; *Ulman et al., 2017*), and the second type captures limb regeneration in *Parhyale hawaiensis* (PH dataset, imaging adapted from *Alwes et al., 2016*), which presents greater challenges for image analysis (see below, *Figure 5—video 1*). For both types of dataset, we find that fewer than 10 annotated nuclei are sufficient to initiate a virtuous cycle of training, prediction, and proofreading, which efficiently yields cell tracks and validated cell lineages in highly dynamic tissues. Interactive cycles of manual annotation, deep learning, and proofreading on ELEPHANT reduce the time required to detect and validate nuclei (*Figure 2B*). On the CE1 dataset, a complete cell lineage was built over 195 timepoints, from scratch, using ELEPHANT's semi-automated workflow (*Figure 4B*). The detection model was trained incrementally starting from sparse annotations (four nuclei) on the first timepoint. On this dataset, linking could be performed using the nearest neighbor algorithm (without flow modeling) and manual proofreading. In this way, we were able to annotate in less than 8 hr a total of 23,829 nuclei (across 195 timepoints), of which ~ 2% were manually annotated (483 nuclei) and the remaining nuclei were collected by validating predictions of the deep-learning model.

Although ELEPHANT works efficiently without prior training, cell tracking can be accelerated by starting from models trained on image data with similar characteristics. To illustrate this, we used nuclear annotations in a separate dataset, CE2, to train a model for detection, which was then applied to CE1. This pre-trained model allowed us to detect nuclei in CE1 much more rapidly and effortlessly

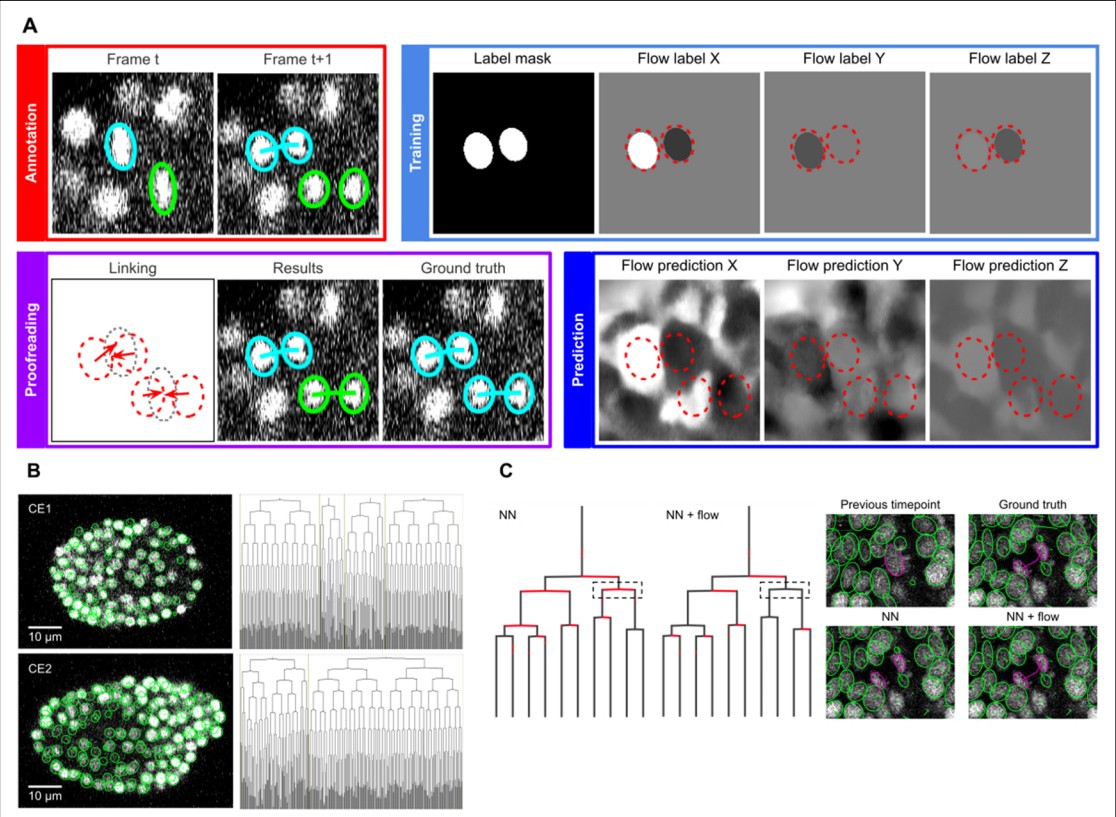

**Figure 4.** ELEPHANT linking workflow. (**A**) Linking workflow, illustrated on the CE1 dataset. Top left: The user annotates links by connecting detected nuclei in successive timepoints; annotated/validated nuclei and links are shown in cyan, non-validated ones in green. Top right: The flow model is trained with optical flow labels coming from annotated nuclei with links (voxels indicated in the label mask), which consist of displacements in X, Y, and Z; greyscale values indicate displacements along a given axis, annotated nuclei with link labels are outlined in red. Bottom right: The trained model generates voxel-wise flow maps for each axis; greyscale values indicate displacements, annotated nuclei are outlined in red. Bottom left: The user validates or rejects the predictions; predicted links are shown in green, predicted and validated links in cyan. (**B**) Tracking results obtained with ELEPHANT. Left panels: Tracked nuclei in the CE1 and CE2 datasets at timepoints 194 and 189, respectively. Representative optical sections are shown with tracked nuclei shown in green; out of focus nuclei are shown as green spots. Right panels: Corresponding lineage trees. (**C**) Comparison of tracking results obtained on the PH dataset, using the nearest neighbor algorithm (NN) with and without optical flow prediction (left panels); linking errors are highlighted in red on the correct lineage tree. The panels on the right focus on the nuclear division that is marked by a dashed line rectangle. Without optical flow prediction, the dividing nuclei (in magenta) are linked incorrectly.

The online version of this article includes the following video and figure supplement(s) for figure 4:

**Figure supplement 1.** 3D U-Net architecture for flow.

**Figure 4—video 1.** ELEPHANT flow predictions in 3D.

https://elifesciences.org/articles/69380/figures#fig4video1

than with an untrained model (*Figure 2B*, blue versus orange curves). For benchmarking, the detection and linkage models trained with the annotations from the CE1 and CE2 lineage trees were then tested on unseen datasets with similar characteristics (without proofreading), as part of the Cell Tracking Challenge (*Maška et al., 2014*; *Ulman et al., 2017*). In this test, our models with assistance of flow-based interpolation (see Materials and methods) outperformed state-of-the-art tracking algorithms (*Magnusson et al., 2015*; *Scherr et al., 2020*) in detection (DET) and tracking (TRA) metrics (*Table 1*). ELEPHANT performs less well in segmentation (SEG), probably due to the use of ellipsoids to approximate nuclear shapes.

The PH dataset presents greater challenges for image analysis, such as larger variations in the shape, intensity, and distribution of nuclei, lower temporal resolution, and more noise (*Figure 5— figure supplement 1*). ELEPHANT has allowed us to grapple with these issues by supporting the continued training of the models through visual feedback from the user (annotation of missed nuclei,

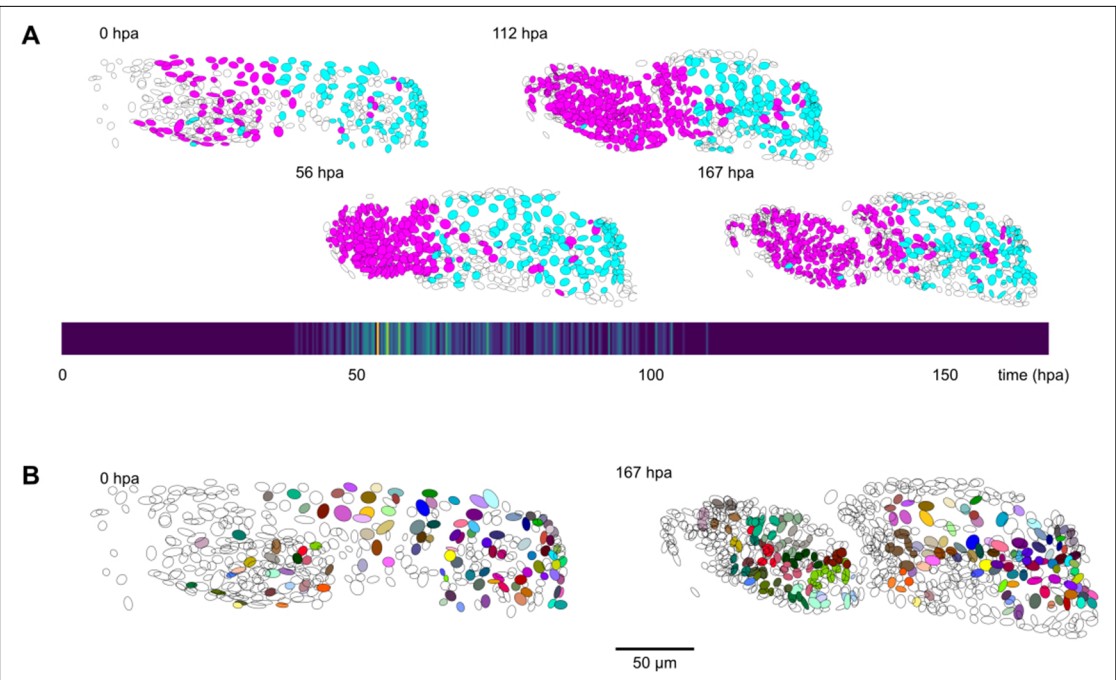

**Figure 5.** Cell lineages tracked during the time course of leg regeneration. (**A**) Spatial and temporal distribution of dividing nuclei in the regenerating leg of *Parhyale* tracked over a 1-week time course (PH dataset), showing that cell proliferation is concentrated at the distal part of the regenerating leg stump and peaks after a period of proliferative quiescence, as described in *Alwes et al., 2016*. Top: Nuclei in lineages that contain at least one division are colored in magenta, nuclei in non-dividing lineages are in cyan, and nuclei in which the division status is undetermined are blank (see Materials and methods). Bottom: Heat map of the temporal distribution of nuclear divisions; hpa, hours post amputation. The number of divisions per 20-min time interval ranges from 0 (purple) to 9 (yellow). (**B**) Fate map of the regenerating leg of *Parhyale*, encompassing 109 fully tracked lineage trees (202 cells at 167 hpa). Each clone is assigned a unique color and contains 1–9 cells at 167 hpa. Partly tracked nuclei are blank. In both panels, the amputation plane (distal end of the limb) is located on the left.

The online version of this article includes the following video and figure supplement(s) for figure 5:

**Figure supplement 1.** Image quality issues in the PH dataset.

**Figure supplement 2.** Complete cell lineage trees in a regenerating leg of *Parhyale*.

**Figure 5—video 1.** Live imaging of *Parhyale* leg regeneration (PH dataset).

https://elifesciences.org/articles/69380/figures#fig5video1

validation and rejection of predictions). Using ELEPHANT, we annotated and validated over 260,000 nuclei in this dataset, across 504 timepoints spanning 168 hr of imaging.

We observed that the conventional nearest neighbor approach was inadequate for linking in the PH dataset, resulting in many errors in the lineage trees (*Figure 4C*). This is likely due to the lower temporal resolution in this dataset (20 min in PH, versus 1–2 min in CE) and the fact that daughter nuclei often show large displacements at the end of mitosis. We trained optical flow using 1,162 validated links collected from 10 timepoints (including 18 links for 9 cell divisions). These sparse annotations were sufficient to generate 3D optical flow predictions for the entire dataset (*Figure 4—video 1*), which significantly improved the linking performance (*Figure 4C*, *Supplementary file 2*): the number of false positive and false negative links decreased by ~57% (from 2093 to 905) and ~32% (from 1991 to 1349), respectively, among a total of 259,071 links.

By applying ELEPHANT's human-in-the-loop semi-automated workflow, we succeeded in reconstructing 109 complete and fully validated cell lineage trees encompassing the duration of leg regeneration in *Parhyale*, each lineage spanning a period of ~1 week (504 timepoints, *Figure 5—figure supplement 2*). Using analysis and visualization modules implemented in Mastodon and ELEPHANT, we could capture the distribution of cell divisions across time and space (*Figure 5A*) and produce a fate map of the regenerating leg of *Parhyale* (*Figure 5B*). This analysis, which would have required several months of manual annotation, was achieved in ~1 month of interactive cell tracking in ELEPHANT,

without prior training. Applying the best performing models to new data could improve tracking efficiency even further.

## Materials and methods
### Image datasets
The PH dataset (dataset li13) was obtained by imaging a regenerating T4 leg of the crustacean *Parhyale hawaiensis*, based on the method described by *Alwes et al., 2016*; *Figure 5—video 1*. The imaging was carried out on a transgenic animal carrying the *Mi(3xP3> DsRed; PhHS> H2B-mRFPRuby)* construct (*Wolff et al., 2018*), in which nuclear-localised mRFPRuby fluorescent protein is expressed in all cells following heat-shock. The leg was amputated at the distal end of the carpus. Following the amputation, continuous live imaging over a period of 1 week was performed on a Zeiss LSM 800 confocal microscope equipped with a Plan-Apochromat 20 x/0.8 M27 objective (Zeiss 420650-9901-000), in a temperature control chamber set to 26 °C. Heat-shocks (45 minutes at 37 °C) were applied 24 hr prior to the amputation, and 65 and 138 hr post-amputation. Every 20 min we recorded a stack of 11 optical sections, with a z step of 2.48 microns. Voxel size (in xyz) was 0.31 × 0.31 x 2.48 microns.

The CE1 and CE2 datasets (*Murray et al., 2008*) and the MDA231 dataset were obtained via the Cell Tracking Challenge (*Ulman et al., 2017*) (datasets Fluo-N3DH-CE and Fluo-C3DL-MDA231). The ORG1 and ORG2 datasets were obtained from *de Medeiros, 2021* and *Kok et al., 2020*, respectively. Additional datasets used to train the generic models (see *Supplementary file 3*) were obtained from the Cell Tracking Challenge (*Ulman et al., 2017*).

### ELEPHANT platform architecture
ELEPHANT implements a client-server architecture (*Figure 1—figure supplement 1*), which can be set up on the same computer or on multiple connected computers. This architecture brings flexibility: allowing the client to run Mastodon (implemented in Java) while the deep learning module is implemented separately using Python, and releasing the client computer from the requirements of high GPU needed to implement deep learning. The client side is implemented by extending Mastodon, a framework for cell tracking built upon the SciJava ecosystem (https://scijava.org/) and is available as a Fiji (*Schindelin et al., 2012*) plugin. Combining the BigDataViewer (*Pietzsch et al., 2015*) with an efficient memory access strategy (https://github.com/mastodon-sc/mastodon/blob/master/doc/trackmate-graph.pdf), Mastodon enables fast and responsive user interaction even for very large datasets. ELEPHANT leverages the functionalities provided by Mastodon, including the functions for manual annotation of nuclei, and extends them by implementing modules for deep learning-based algorithms.

The server side is built using an integrated system of a deep learning library (PyTorch *Paszke et al., 2019*), tools for tensor computing and image processing (NumPy *Harris et al., 2020*), SciPy (*Virtanen et al., 2020*), Scikit Image (*van der Walt et al., 2014*), and web technologies (Nginx, uWSGI, Flask). The client and the server communicate by Hypertext Transfer Protocol (HTTP) and JavaScript Object Notation (JSON). To reduce the amount of data exchanged between the client and the server, the image data is duplicated and stored in an appropriate format on each side. An in-memory data structure (Redis) is used to organize the priorities of the HTTP requests sent by the client. A message queue (RabbitMQ) is used to notify the client that the model is updated during training. The client software is available as an extension on Fiji (https://github.com/elephant-track/elephant-client). The server environment is provided as a Docker container to ensure easy and reproducible deployment (https://github.com/elephant-track/elephant-server). The server can also be set up with Google Colab in case the user does not have access to a computer that satisfies the system requirements.

### Computer setup and specifications
In this study, we set up the client and the server on the same desktop computer (Dell Alienware Aurora R6) with the following specifications: Intel Core i7-8700K CPU @3.70 GHz, Ubuntu 18.04, 4 × 16 GB DDR4 2,666 MHz RAM, NVIDIA GeForce GTX 1080 Ti 11 GB GDDR5X (used for deep learning), NVIDIA GeForce GTX 1650 4 GB GDDR5, 256 GB SSD and 2 TB HDD. System requirements for the client and the server are summarized in the user manual (https://elephant-track.github.io/).

## Dataset preparation

Images were loaded in the BigDataViewer (BDV, *Pietzsch et al., 2015*) format on the client software. The CE1, CE2, ORG1, ORG2, and MDA231 datasets were converted to the BDV format using the BigDataViewer Fiji plugin (https://imagej.net/BigDataViewer) without any preprocessing. Because the PH dataset showed non-negligible variations in intensity during long-term imaging, the original 16-bit images were intensity normalized per timepoint before conversion to the BDV format, for better visualization on Mastodon. In this normalization, the intensity values were re-scaled so that the minimum and maximum values at each timepoint become 0 and 65535, respectively. The PH dataset also showed 3D drifts due to heat-shocks. The xy drifts were corrected using an extended version of image alignment tool (*Tseng et al., 2011*) working as an ImageJ (*Schneider et al., 2012*) plugin, where the maximum intensity projection images were used to estimate the xy displacements, subsequently applied to the whole image stack (https://github.com/elephant-track/align-slices3d). The z drifts were corrected manually by visual inspection using Fiji.

On the server, images, annotation labels and outputs were stored in the Zarr format, allowing fast read/write access to subsets of image data using chunk arrays (*Moore et al., 2021*). At the beginning of the analysis, these data were prepared using a custom Python script that converts the original image data from HDF5 to Zarr and creates empty Zarr files for storing annotation labels and outputs (https://github.com/elephant-track/elephant-server). This conversion can also be performed from the client application. Generally, HDF5 is slower in writing data than Zarr, especially in parallelization, while they show comparable reading speeds (*Moore et al., 2021*).

On the server, the image data are stored in unsigned 8-bit or unsigned 16-bit format, keeping the original image format. At the beginning of processing on the server, the image data are automatically converted to a 32-bit float and their intensity is normalized at each timepoint such that the minimum and maximum values become 0 and 1.

## Algorithm for detection

Detection of nuclei relies on three components: (i) a U-Net CNN that outputs probability maps for *nucleus center*, *nucleus periphery*, and *background*, (ii) a post-processing workflow that extracts *nucleus center* voxels from the probability maps, (iii) a module that reconstructs nuclei instances as ellipsoids. We designed a variation of 3D U-Net (*Cicek et al., 2016*) as illustrated in *Figure 2—figure supplement 1*. In both encoder and decoder paths, repeated sets of 3D convolution, ReLU activation (*Nair and Hinton, 2010*) and Group Normalization (*Wu and He, 2020*) are employed. Max pooling in 3D is used for successive downsampling in the encoder path, in each step reducing the size to half the input size (in case of anisotropy, maintaining the z dimension until the image becomes nearly isotropic). Conversely, in the decoder path, upsampling with nearest-neighbor interpolation is applied to make the dimensions the same as in the corresponding intermediate layers in the encoder path. As a result, we built a CNN with 5,887,011 trainable parameters. The weights are initialized with the Kaiming fan-in algorithm (*He et al., 2015a*) and the biases are initialized to zero for each convolution layer. For each group normalization layer, the number of groups is set as the smallest value between 32 and the number of output channels, and the weights and biases are respectively initialized to one and zero. When starting to train from scratch, the CNN is trained using the cropped out 3D volumes from the original image prior to training with annotations. In this prior training phase, a loss function $L_{prior}$ is used that penalizes the addition of the following two mean absolute differences (MADs): (i) *nucleus center* probabilities $c_i$ and the [0, 1] normalized intensity of the original image $y_i$, (ii) *background* probabilities $b_i$ and the [0, 1] normalized intensity of the intensity-inverted image $1 - y_i$ where $i$ stands for the voxel index of an input volume with $n$ voxels $i \in V := \{1, 2, \ldots, n\}$.

$$L_{prior} = \frac{1}{n} \sum_{i=1}^{n} |y_i - c_i| + \frac{1}{n} \sum_{i=1}^{n} |(1 - y_i) - b_i|$$

The prior training is performed on three cropped out 3D volumes generated from the 4D datasets, where the timepoints are randomly picked, and the volumes are randomly cropped with random scaling in the range (0.8, 1.2). The training is iterated for three epochs with decreasing learning rates (0.01, 0.001, and 0.0001, in this order) with the Adam optimizer (*Kingma and Ba, 2014*). The prior training can be completed in ~20 s for each dataset.

Training with sparse annotations is performed in the following steps. First, the client application extracts the timepoint, 3D coordinates and covariances representing ellipsoids of all the annotated

nuclei in the specified time range. Subsequently, these data, combined with user-specified parameters for training, are embedded in JSON and sent to the server in an HTTP request. On the server side, training labels are generated from the received information by rendering *nucleus center*, *nucleus periphery*, *background* and unlabelled voxels with distinct values. The *background* labels are generated either by explicit manual annotation or intensity thresholding, where the threshold value is specified by the user, resulting in the label images as shown in **Figure 2A**. To render ellipsoids in the anisotropic dimension, we extended the draw module in the scikit-image library (**van der Walt et al., 2014**) (https://github.com/elephant-track/elephant-server). Training of the CNN is performed using the image volumes as input and the generated labels as target with a loss function $L_{vclass}$ that consists of three terms: (i) a class-weighted negative log-likelihood (NLL) loss, (ii) a term computed as one minus the dice coefficient for the *nucleus center* voxels, and (iii) a term that penalizes the roughness of the *nucleus center* areas. We used the empirically-defined class weights **wc** for the NLL loss: *nucleus center* = 10, *nucleus periphery* = 10, *background* = 1; the unlabelled voxels are ignored. The first two terms accept different weights for the true annotations *wt* (i.e. true positive and true negative) and the false annotations *wf* (i.e. false positive and false negative). The third term is defined as the MAD between the voxel-wise probabilities for *nucleus center* and its smoothed representations, which are calculated by the Gaussian filter with downsampling (*Down*) and upsampling (*Up*). Let $i$ stand for the voxel index of an input volume with $n$ voxels $i \in V := \{1, 2, ..., n\}$, $x_i$ for the input voxel value, $\mathbf{h}_i$ for the output from the CNN before the last activation layer for the three classes, $y_i \in Y := \{1, 2, 3\}$ for the voxel class label (1: *nucleus center*, 2: *nucleus periphery*, 3: *background*, respectively), and $z_i \in Z := \{ true, false, unlabeled \}$ for the voxel annotation label. We define the following subsets: the voxel index with true labels $T = \{ i \mid i \in V, z_i = false \}$, with false labels $F = \{ i \mid i \in V, z_i = false \}$, and the *nucleus center* $C = \{ i \mid i \in V, y_i = 1 \}$. In the calculation of the $L_{dice}$, a constant $\epsilon = 0.000001$ is used to prevent zero division. Using these components and the empirically-defined weights for each loss term ($\alpha = 1, \beta = 5, \gamma = 1$), we defined the $L_{vclass}$ as below.

$$L_{vclass} = \alpha L_{nll} + \beta L_{dice} + \gamma L_{smooth}$$

$$L_{nll} = wt \frac{\sum_{i \in T}\left(NLL\left(\mathbf{h}_i, y_i\right) \cdot wc[y_i]\right)}{\sum_{i \in T} wc[y_i]} + wf \frac{\sum_{i \in F}\left(NLL\left(\mathbf{h}_i, y_i\right) \cdot wc[y_i]\right)}{\sum_{i \in F} wc[y_i]}$$

$$L_{dice} = wt \left( 1 - \frac{2\sum_{i \in T}(Prob(\mathbf{h}_i, 1) \cdot Onehot(y_i, 1))}{max\left(\sum_{i \in T}\left(\left(Prob\left(\mathbf{h}_i, 1\right)\right)^2 + \left(Onehot\left(y_i, 1\right)\right)^2\right), \epsilon\right)} \right)$$

$$+ wf \left( 1 - \frac{2\sum_{i \in F}(Prob(\mathbf{h}_i, 1) \cdot Onehot(y_i, 1))}{max\left(\sum_{i \in F}\left(\left(Prob\left(\mathbf{h}_i, 1\right)\right)^2 + \left(Onehot\left(y_i, 1\right)\right)^2\right), \epsilon\right)} \right)$$

$$L_{smooth} = \frac{1}{n}\sum_{i \in V}|Prob\left(\mathbf{h}_i, 1\right) - Up\left(Down\left(Prob\left(\mathbf{h}_i, 1\right)\right)\right)|$$

$$Prob\left(\mathbf{h}, c\right) = \frac{exp\left(\mathbf{h}[c]\right)}{\sum_{j=1}^{3} exp\left(\mathbf{h}[j]\right)}$$

$$NLL\left(\mathbf{h}, y\right) = -\log\left(Prob\left(\mathbf{h}, y\right)\right)$$

$$Onehot\left(y_i, c\right) = \begin{cases} 0 & \left(y_i \neq c\right) \\ 1 & \left(y_i = c\right) \end{cases}$$

In the analyses shown in **Figure 3** and its supplements 1 and 2, the following loss functions are updated to make them more robust using normalization, which are employed in the current version of software.

$$L_{vclass} = \frac{\alpha L_{nll} + \beta L_{dice} + \gamma L_{smooth}}{\alpha + \beta + \gamma}$$

$$L_{nll} = \frac{n(T)wt}{n(T)wt + n(F)wf} \frac{\sum_{i \in T}\left(NLL\left(\mathbf{h}_i, y_i\right) \cdot wc[y_i]\right)}{\sum_{i \in T} wc[y_i]} + \frac{n(F)wf}{n(T)wt + n(F)wf} \frac{\sum_{i \in F}\left(NLL\left(\mathbf{h}_i, y_i\right) \cdot wc[y_i]\right)}{\sum_{i \in F} wc[y_i]}$$

$$L_{dice} = \frac{n(T)wt}{n(T)wt + n(F)wf} \left( 1 - \frac{2\sum_{i \in T}(Prob(\mathbf{h}_i, 1) \cdot Onehot(y_i, 1))}{max\left(\sum_{i \in T}\left(\left(Prob\left(\mathbf{h}_i, 1\right)\right)^2 + \left(Onehot\left(y_i, 1\right)\right)^2\right), \epsilon\right)} \right)$$

$$+ \frac{n(F)wf}{n(T)wt + n(F)wf} \left( 1 - \frac{2\sum_{i \in F}(Prob(\mathbf{h}_i, 1) \cdot Onehot(y_i, 1))}{max\left(\sum_{i \in F}\left(\left(Prob\left(\mathbf{h}_i, 1\right)\right)^2 + \left(Onehot\left(y_i, 1\right)\right)^2\right), \epsilon\right)} \right)$$

Training of the CNN is performed on the image volumes generated from the 4D datasets, where the volumes are randomly cropped with/without random scaling, random contrast, random flip and random rotation, which are specified at runtime. There are two modes for training: (i) an interactive mode that trains a model incrementally, as the annotations are updated, and (ii) a batch mode that trains a model with a fixed set of annotations. In the interactive training mode, sparse annotations in a given timepoint are used to generate crops of image and label volumes, with which training is performed using the Adam optimizer with a learning rate specified by the user. In the batch training mode, a set of crops of image and label volumes per timepoint is generated each iteration, with which training is performed for a number of epochs specified by the user (ranging from 1 to 1000) using the Adam optimizer with the specified learning rates. In the prediction phase, the input volume can be cropped into several blocks with smaller size than the original size to make the volume can be cropped into several blocks with smaller size than the original size to make the input data compatible with available GPU memory. To stitch the output blocks together, the overlapping regions are seamlessly blended by weighted linear blending.

In post-processing for the CNN output, voxel-wise probabilities for *nucleus center* class are denoised by subtracting edges of *background* class that are calculated with the Gaussian filter and the Prewitt operation for each z-slice. After denoising, the voxels with *nucleus center* probabilities greater than a user defined value are thresholded and extracted as connected components, which are then represented as ellipsoids (from their central moments). These ellipsoids representing the *nucleus center* regions are enlarged so that they cover the original nucleus size (without excluding its periphery). The ellipsoids with radii smaller than $r_{min}$ are removed and the radii are clamped to $r_{max}$ specified by the user, generating a list of center positions and covariances that can be used to reconstruct the nuclei. On the client application, the detection results are converted to Mastodon spots and rendered on the BDV view, where the existing and predicted nuclei are tagged based on their status: labelled as 'true positive' (positive and predicted), 'false negative' (positive and not predicted), 'true negative' (negative and not predicted), 'false positive' (negative and predicted), and 'non-validated' (newly predicted). These labels can be visualized when running ELEPHANT in the advanced color mode (in basic color mode true positives and false negatives are visualized as 'accepted' and false positives and true negatives as 'rejected'). If more than one nucleus is predicted within a user-specified threshold $d_{sup}$, the one with human annotation is given priority, followed by the one with the largest volume.

## Algorithm for linking

Linking of nuclei relies on two components: (i) estimation of the positions of nuclei at the previous timepoint by optical flow estimation using deep learning, which is skipped in the case of the nearest neighbor algorithm without flow support, (ii) association of nuclei based on the nearest neighbor algorithm. We designed a variation of 3D U-Net for flow estimation as illustrated in *Figure 4—figure supplement 1*. In the encoder path, the residual blocks (*He et al., 2015b*) with 3D convolution and LeakyReLU (*Maas et al., 2013*) activation are applied, in which the outputs are divided by two after the sum operation to keep the consistency of the scale of values. In the decoder path, repeated sets of 3D convolution and LeakyReLU activation are employed. Downsampling and upsampling are applied as described for the detection model. Tanh activation is used as a final activation layer. As a result, we built a CNN with 5,928,051 trainable parameters. The weights and biases for convolution layers are initialized as described for the detection model. Training of the flow model with sparse annotations is performed in a similar way as for the detection model. First, on the client application, for each annotated link, which connects the source and target nuclei, the following information gets extracted: the timepoint, the backward displacements in each of the three dimensions, and the properties of the target nucleus (3D coordinates and covariances). Subsequently, these data, combined with parameters for training, are embedded in JSON and sent to the server in an HTTP request. On the server side, flow labels are generated from the received information by rendering backward displacements for each target nucleus in each of three dimensions, where the displacements are scaled to fit the range (–1, 1). In this study, we used fixed scaling factors (1/80, 1/80, 1/10) for each dimension, but they can be customized to the target dataset. Foreground masks are generated at the same time to ignore unlabelled voxels during loss calculation. Ellipsoid rendering is performed as described for the detection training. Training of the CNN for flow estimation is performed using the two consecutive image volumes $(I_{t-1}, I_t)$ as input, and the generated label as target. A loss function $L_{flow}$ is defined with the

following three terms; (i) a dimension-weighted MAD between the CNN outputs and the flow labels, (ii) a term computed as one minus the structural similarity (SSIM) (*Wang et al., 2004*) of $I_{t-1}$ and $\tilde{I}_t$, where the estimated flow is applied to $I_t$ (*Ilg et al., 2017*), (iii) a term penalizing the roughness of the CNN outputs. Let $i$ stand for the voxel index of an input volume with $n$ voxels $i \in V := \{1, 2, \ldots, n\}$, $x_i$ for the input voxel value, $\mathbf{\hat{y}}$ for the output of the CNN, $\mathbf{y}$ for the flow label, $m \in M \subset V$ for the index of the annotated voxels, $d \in D := \{0, 1, 2\}$ for the dimension index for three dimensions and $\mathbf{wd}$ for the dimension weights. In the SSIM calculation, we defined a function *Gauss* as a 3D Gaussian filter with the window size 7 × 7 x 3 and standard deviation of 1.5. Using these components and the empirically defined weights for each loss term ($\alpha = 1, \beta = 0.0001, \gamma = 0.0001$), we defined the $L_{flow}$ as below.

$$L_{flow} = \alpha L_{mad} + \beta L_{ssim} + \gamma L_{smooth}$$

$$L_{mad} = \frac{1}{n} \sum_{d \in D} wd_d \sum_{m \in M} |\mathbf{y}_{md} - \mathbf{\hat{y}}_{md}|$$

$$L_{ssim} = 1 - SSIM\left(I_{t-1}, \hat{I}_t\right)$$

$$L_{smooth} = \frac{1}{3n} \sum_{i \in V} \sum_{d \in D} |\mathbf{\hat{y}}_{id} - Up\left(Down\left(\mathbf{\hat{y}}_{id}\right)\right)|$$

$$\mu_{I_1} = Gauss\left(I_1\right), \mu_{I_2} = Gauss\left(I_2\right)$$

$$\sigma_{I_1}^2 = Gauss\left(I_1^2\right) - \mu_{I_1}^2, \sigma_{I_2}^2 = Gauss\left(I_2^2\right) - \mu_{I_2}^2, \sigma_{I_1 I_2} = Gauss\left(I_1 I_2\right) - \mu_{I_1}\mu_{I_2}$$

$$SSIM\left(I_1, I_2\right) = \frac{\left(2\mu_{I_1}\mu_{I_2} + C_1\right)\left(2\sigma_{I_1 I_2} + C_2\right)}{\left(\mu_{I_1}^2 + \mu_{I_2}^2 + C_1\right)\left(\sigma_{I_1}^2 + \sigma_{I_2}^2 + C_2\right)}$$

, where $C_1 = 0.0001$ and $C_2 = 0.0009$. In the current version of software, the following loss functions are updated to make them more robust using normalization.

$$L_{flow} = \frac{\alpha L_{mad} + \beta L_{ssim} + \gamma L_{smooth}}{\alpha + \beta + \gamma} \quad (\alpha = 1, \beta = 0.01, \gamma = 0.01)$$

$$L_{mod} = \frac{1}{n(M)} \sum_{d \in D} \frac{wd_d}{\sum_{d \in D} wd_d} \sum_{m \in M} |\mathbf{y}_{md} - \mathbf{\hat{y}}_{md}|$$

The training is performed on the image volumes generated from the 4D datasets, where the sets of two consecutive images and corresponding flow labels are randomly cropped with/without random scaling and random rotation, which are specified at runtime. The training is performed for a fixed number of epochs using the Adam optimizer and with learning rates specified by the user, generating a set of images and labels for each timepoint in each epoch. The CNN outputs are rescaled to the original physical scale and used to calculate the estimated coordinate of each nucleus center at the previous timepoint. Let $K \subset V$ stands for a subset of voxel index of a nucleus and $\mathbf{p}$ for its center coordinate. Using the output of the CNN $\mathbf{\hat{y}}$ and the scaling factor $s$, the estimated coordinate at the previous timepoint $\mathbf{\hat{p}}$ is calculated.

$$\mathbf{\hat{p}} = \mathbf{p} + \frac{s}{n(K)} \sum_{k \in K} \mathbf{\hat{y}}_k$$

These estimated coordinates are subsequently used to find the parent of the nucleus at the previous timepoint by the nearest neighbor algorithm (a similar concept was introduced for 2D phase contrast microscopy data; *Hayashida and Bise, 2019*; *Hayashida et al., 2020*). The pairs with a distance smaller than $d_{search}$ are considered as link candidates, where the closer the Euclidean distance between the two points, the higher their priority of being the correct link. Each nucleus accepts either one or two links, determined by the estimated displacements and actual distances. Briefly, given that a single nucleus has two possible links, it can accept both if at least one of the estimated displacements is larger than the threshold $d_{disp}$ or both distances are smaller than the threshold $d_{dist}$. In this study, we used ad hoc thresholds $d_{disp} = 1.0$ and $d_{dist} = 1.0$. If there are competing links beyond the allowed maximum of two links, the links with smaller $d_{disp}$ are adopted and the remaining nucleus looks for the next closest nucleus up to $N_{max}$ neighbors. The links are generated by repeating the above procedure until all the nuclei get linked or the iteration count reaches to five. We optionally implement an interpolation algorithm, in which each orphan nucleus tries to find its source up to $T_{max}$ timepoints back and is linked with a nucleus at the estimated coordinate based on the flow prediction, interpolating the points in between.

## Preparation of generic pre-trained models and fine-tuning with sparse annotations (Figure 3)

The generic pre-trained models for each dataset were trained on the datasets summarized in *Supplementary file 3*. Training of the detection models was performed with volumes of 384 × 384 x 16 voxels or smaller, which were generated by preprocessing with random flip in each dimension, random scaling in the range (0.5, 2), random cropping and random contrast in the range (0.5, 1). In the label generation step, the center ratio was set to 0.4 and the background threshold was set to 1 (i.e. all voxels without manual annotations are background) for the Cell Tracking Challenge datasets (Fluo-C3DH-A549, Fluo-C3DH-H157, Fluo-C3DL-MDA231, Fluo-N3DH-CE and Fluo-N3DH-CHO), and to 0.03 for the PH dataset. The labels for the Cell Tracking Challenge datasets were automatically generated from the silver-standard corpus (silver truth). We trained the models for up to 200 epochs starting from scratch using the Adam optimizer with the learning rate of $5 \times 10^{-3}$, where each epoch contained randomly selected 10 pre-processed volumes from each dataset. Validation was performed after each epoch using randomly selected five timepoints from each dataset. In the validation phase, image volumes were fed into the model using blocks with size 512 × 512 x 24 or smaller, and the outputs were stitched together to reconstruct the whole volume. For each condition, the model with the highest score in the validation data was finally adopted. At the start of each epoch, the model parameters were set to the ones that had previously produced the highest scores on the validation data. The parameters for training and validation are summarized in *Supplementary file 5*. Fine-tuning of the model was performed as follows: (i) 3–10 sparse annotations were added at the points where the pre-trained model failed in detection (*Figure 3* middle), (ii) we trained the models for 10 epochs starting from the pre-trained model parameters with volumes of 384 × 384 x 16 voxels or smaller. These volumes were generated by preprocessing with random flip in each dimension, random scaling in the range (0.5, 2), random cropping and random contrast in the range (0.5, 1), using the Adam optimizer with the learning rate of 0.01 or 0.001, where each epoch contained five randomly cropped volumes. The pre-trained model and the fine-tuned model were applied to each dataset with parameters summarized in *Supplementary file 6*. The evaluation scores were calculated based on the detection criterion of ELEPHANT, which recognizes that a prediction is correct if the distance between the prediction and the manual annotation is less than $d_{sup}$.

## Comparison between ELEPHANT and StarDist3D (Figure 3 – Supplement 1)

The ELEPHANT detection model is the same as the one used in *Figure 3—figure supplement 2*. For training of the StarDist3D (*Weigert et al., 2020*) segmentation model, a single volume of the CE1 dataset (timepoint = 100) with the fully labelled instance segmentation annotations (93 nuclei) was used for training, and another volume of the CE1 dataset (timepoint = 101) with the fully labelled instance segmentation annotations (95 nuclei) was used for validation during training. The instance segmentation annotations were taken from the silver-standard corpus (silver truth) in the Cell Tracking Challenge. Training of the StarDist3D model was performed with the parameters summarized in *Supplementary file 4*. The model with the best performance in the validation data was selected for comparison. The trained ELEPHANT and StarDist3D models were applied to a single volume (timepoint = 100) of the CE2 dataset to generate the output for comparison (*Figure 3—figure supplement 1*). The DET scores were calculated by the evaluation software provided by the Cell Tracking Challenge using the gold-standard corpus (gold truth).

## Evaluation of overfitting of the detection model (Figure 3 – Supplement 2)

A single volume of the CE1 dataset (timepoint = 100) with sparse annotations (10 nuclei) was used for training, and a single volume of the fully labelled CE2 dataset (timepoint = 100) was used for validation. Training of the detection model was performed using 384 × 384 x 16 cropped-out image volumes generated by preprocessing with random flip in each dimension, random scaling in the range (0.5, 2), random cropping and random contrast in the range (0.5, 1). In the label generation step, the center ratio was set to 0.4 and the background threshold was set to 0 (i.e. all voxels without manual annotations are ignored). We trained a model for 500 epochs starting from scratch using the Adam optimizer with the learning rate of $5 \times 10^{-4}$, where each epoch contained five pre-processed volumes.

Training and validation losses were recorded at the end of each epoch (*Figure 3—figure supplement 2*, bottom left). The detection model trained for 500 epochs was tested on the CE1 dataset (partially-seen data; *Figure 3—figure supplement 2*, top right) and the CE2 dataset (unseen data; *Figure 3—figure supplement 2*, bottom right). In the prediction phase, the input volumes were cropped into 2 × 2 x 2 blocks with size 544 × 384 x 28 for CE1 or 544 × 384 x 24 for CE2, and stitched together to reconstruct the whole image of 708 × 512 x 35 for CE1 or 712 × 512 x 31 for CE2. In the postprocessing of the prediction for detection, a threshold for the nucleus center probabilities were set to 0.5, and $r_{min}$, $r_{max}$ and $d_{sup}$ were set to 0.5 μm, 3 μm, and 2 μm respectively. The evaluation scores were calculated in the same way as described in the previous section.

### Detection and tracking in the CE datasets (Figures 2 and 4)

On the CE1 and CE2 datasets, training of detection and flow models was performed with volumes of 384 × 384 x 16 voxels that were generated by preprocessing with random scaling in the range (0.5, 2) and random cropping. For training of a detection model, preprocessing with random contrast in the range (0.5, 1) was also applied. In the label generation step, the center ratio was set to 0.3 and the background threshold was set to 0.1 and 1 (i.e. all voxels without manual annotations are background). In the interactive training of detection models, 10 labelled cropped out volumes were generated per iteration, with which training was performed using the Adam optimizer with a learning rate between $5 \times 10^{-5}$ and $5 \times 10^{-6}$. In the batch training of detection models, training was performed for 100 epochs using the Adam optimizer with learning rates of $5 \times 10^{-5}$. In the training of a flow model, training was performed for 100 epochs using the Adam optimizer with learning rates of $5 \times 10^{-5}$ for the first 50 epochs and $5 \times 10^{-6}$ for the last 50 epochs. *wt* and *wf* were set to 1 and 5, respectively, and **wd** was set to (1/3, 1/3, 1/3). In the prediction phase, the input volumes were cropped into 2 × 2 x 2 blocks with size 544 × 384 x 28 for CE1 or 544 × 384 x 24 for CE2, and stitched together to reconstruct the whole image of 708 × 512 x 35 for CE1 or 712 × 512 x 31 for CE2. In the preprocessing of the prediction for detection, we corrected the uneven background levels across the z-slices by shifting the slice-wise median value to the volume-wise median value. In the postprocessing of the prediction for detection, a threshold for the *nucleus center* probabilities were set to 0.3, and $r_{min}$ and $r_{max}$, $d_{sup}$ were set to 1 μm, 3 μm and 1 μm, respectively. In the nearest-neighbor linking with/without flow prediction, $d_{search}$ was set to 5 μm and $N_{max}$ was set to 3. In the results submitted to the Cell Tracking Challenge (*Table 1*), the suppression of detections with $d_{sup}$ was not applied, and the linking was performed by the nearest-neighbor linking with flow support and an optional interpolation module, where $T_{max}$ was set to 5.

### Detection and tracking in the PH dataset

On the PH dataset, training of detection and flow models was performed with volumes of 384 × 384 x 12 voxels generated by preprocessing with random rotation in the range of ±180 degrees and random cropping. For training a detection model, preprocessing with random contrast in the range (0.5, 1) was also applied. In the label generation step, the center ratio was set to 0.3, and the background threshold was set to 0.03. In the interactive training of a detection model, 10 crops of image and label volumes were generated per iteration, with which training was performed using the Adam optimizer with a learning rate between $5 \times 10^{-5}$ and $5 \times 10^{-6}$. In the batch training of a detection model, training was performed for 100 epochs using the Adam optimizer with learning rates of $5 \times 10^{-5}$. In the training of a flow model, training was performed for 100 epochs using the Adam optimizer with learning rates of $5 \times 10^{-5}$ for the first 50 epochs and $5 \times 10^{-6}$ for the last 50 epochs. *wt* and *wf* were set to 1 and 3, respectively, and **wd** was set to (1, 1, 8). In the prediction phase, the input volumes were fed into the CNNs without cropping or further preprocessing. In the postprocessing of the prediction for detection, a threshold for the *nucleus center* probabilities were set to 0.3, and $r_{min}$, $r_{max}$ and $d_{sup}$ were set to 1 μm, 3 μm and 5 μm respectively. In the nearest-neighbor linking with/without flow prediction, $d_{search}$ was set to 5 μm and $N_{max}$ was set to 3.

### Analysis of CE and PH datasets

On the CE1 and CE2 datasets, the detection and link annotations were made starting from timepoint 0 and proceeding forward until timepoints 194 (CE1) and 189 (CE2), respectively. In the CE1 dataset, the detection was made from scratch, based on manual annotation and incremental training, and the

linking was performed by the nearest neighbor algorithm without flow prediction. After completing annotation from timepoint 0 to 194 on the CE1 dataset, the detection and flow models were trained by the batch mode with the fully labelled annotations. In the CE2 dataset, the detection was performed in a similar way as for CE1, by extending the model trained with CE1, and the linking was performed by the nearest neighbor algorithm with flow support using the pre-trained model followed by proof-reading. Incremental training of the detection model was performed when there were annotations from nuclei that were not properly predicted.

On the PH dataset, the annotations were made by iterating the semi-automated workflow. In general, the nuclei with high signal-to-noise ratio (SNR) were annotated early, while the nuclei with low SNR were annotated in a later phase. The detection model was updated frequently to fit the characteristics of each region and timepoint being annotated, while the flow model was updated less frequently. The CE1 dataset was used to evaluate the speed of detection and validation (*Figure 2B*). All workflows started at timepoint 0 and proceeded forward in time, adding and/or validating all the nuclei found in each timepoint. To evaluate the manual workflow, we annotated nuclei using hotkeys that facilitate the annotation of a given nucleus at successive timepoints. To evaluate the ELEPHANT from scratch workflow, we performed prediction with the latest model, followed by proof-reading, including add, modify, or delete operations, and incremental training. At each timepoint, the model was updated with the new annotations added manually or by proofreading. To evaluate the ELEPHANT pre-trained workflow, we performed predictions with a model trained on the CE2 dataset, followed by proofreading without additional training. The numbers of validated nuclei associated with time were counted from the log data. We measured the counts over 30 min after the start of each workflow and plotted them in *Figure 2B*.

To compare the linking performances (*Figure 4C*), we trained the flow model with 1,162 validated links, including 18 links corresponding to 9 cell divisions, from 108 lineage trees collected between timepoints 150 and 159. It took around 30 hr to train the flow model from scratch using these links. Starting from a pre-trained model, the training time can be decreased to a few minutes, providing a major increase in speed compared with training from scratch (*Supplementary file 2*).

The results shown in A and B*Figure 5A, B* were generated based on the tracking results with 260,600 validated nuclei and 259,071 validated links. In the analysis for *Figure 5A*, nuclei were categorised as dividing or non-dividing depending on whether the lineages to which they belong contain at least one cell division or not during the period of cell proliferation (timepoints 100–350). Nuclei that did not meet these criteria were left undetermined. For *Figure 5B*, the complete lineages of 109 nuclei were tracked through the entire duration of the recording, from 0 to 167 hr post-amputation, with no missing links.

## Evaluation of cell tracking performance

We submitted our results and executable software to the Cell Tracking Challenge organizers, who evaluated our algorithm's performance, validated its reproducibility using the executable software that we submitted, and provided us with the scores. The details of the detection accuracy (DET), tracking accuracy (TRA), and segmentation accuracy (SEG) metrics can be found in the original paper (*Matula et al., 2015*) and the website (http://celltrackingchallenge.net/evaluation-methodology/). Briefly, the DET score evaluates how many *split*, *delete,* and *add* operations are required to achieve the ground truth starting from the predicted nuclei, reflecting the accuracy of detection; the TRA score evaluates how many *split*, *delete,* and *add* operations for nuclei, and *delete*, *add,* and *alter the semantics* operations for links are required to reconstruct the ground truth lineage trees from the predicted lineage trees, reflecting the accuracy of linking; the SEG score evaluates the overlap of the detected ellipsoids with fully segmented nuclei, reflecting the precision of nucleus segmentation. All three scores range from 0 (poorest) to 1 (best).

## Data availability

The CE1, CE2, and MDA231 datasets are available from the Cell Tracking Challenge website: http://celltrackingchallenge.net/3d-datasets/. The ORG1 and ORG2 datasets were obtained from *de Medeiros, 2021* and *Kok et al., 2020*, respectively. The following files are available at https://doi.org/10.5281/zenodo.4630933: (i) the tracking results shown in *Figure 4B*, (ii) the PH dataset and its tracking results, and (iii) deep-learning model parameters for the CE and PH datasets.

## Code availability

The source code for the ELEPHANT client is available at https://github.com/elephant-track/elephant-client (*Sugawara, 2021c* copy archived at swh:1:rev:449f9ff8ad17ce75f355e18f815653ec0aa4bbb8), for the ELEPHANT server at https://github.com/elephant-track/elephant-server (*Sugawara, 2021a* copy archived at swh:1:rev:8935febdbcb2e2d6ba2220ca139e765db44e6458), and for the Align Slices 3D + t extension ImageJ plugin at https://github.com/elephant-track/align-slices3d (*Sugawara, 2021b* copy archived at swh:1:rev:36c6cb6ccb7e308f9349ec26294d408c35be1ed7). The user manual for ELEPHANT is available at https://elephant-trackgithub.io/.

## Acknowledgements

We are grateful to Anna Kreshuk and Constantin Pape for training in machine learning, to Jean-Yves Tinevez (Image Analysis Hub, Institut Pasteur) and Tobias Pietzsch for support in developing ELEPHANT as a Mastodon plugin, to the NEUBIAS community for feedback on the software, to Martin Maška, Michal Kozubek and Carlos Ortiz de Solórzano for support in our submission to the Cell Tracking Challenge, and to Christian Tischer and Sebastian Tosi for extensive feedback on the software and manuscript. We thank Carlos Ortiz de Solórzano, Bob Waterston, Jeroen van Zon, Gustavo de Madeiros and Prisca Liberali for sharing image and cell tracking data used to test ELEPHANT. We also thank Jan Funke, Carsten Wolff, Martin Weigert, Jean-Yves Tinevez, Philipp Keller, Irepan Salvador-Martínez, Severine Urdy, and Mathilde Paris for comments on the manuscript. This research was supported by the European Research Council, under the European Union Horizon 2020 programme, grant ERC-2015-AdG #694918; ÇÇ was supported by a doctoral fellowship from Boehringer Ingelheim Fonds.

## Additional information

### Competing interests

Ko Sugawara: KS is employed part-time by LPIXEL Inc. The other author declares that no competing interests exist.

### Funding

| Funder | Grant reference number | Author |
|---|---|---|
| H2020 European Research Council | ERC-2015-AdG #694918 | Ko Sugawara Michalis Averof |
| Boehringer Ingelheim Fonds | | Çağrı Çevrim |

The funders had no role in study design, data collection and interpretation, or the decision to submit the work for publication.

### Author contributions

Ko Sugawara, Conceptualization, Investigation, Methodology, Software, Validation, Writing – original draft, Writing – review and editing; Çağrı Çevrim, Acquired and annotated imaging data, Validation, Writing – review and editing; Michalis Averof, Conceptualization, Supervision, Writing – original draft, Writing – review and editing

### Author ORCIDs

Ko Sugawara ⬩ http://orcid.org/0000-0002-1392-9340
Çağrı Çevrim ⬩ http://orcid.org/0000-0002-4720-7944
Michalis Averof ⬩ http://orcid.org/0000-0002-6803-7251

### Decision letter and Author response

Decision letter https://doi.org/10.7554/eLife.69380.sa1
Author response https://doi.org/10.7554/eLife.69380.sa2

# Additional files

## Supplementary files

• Supplementary file 1. Processing speed of the detection model Processing speed of the deep learning model for the detection of nuclei, applied to three datasets. The training speed is affected by the distribution of annotations because the algorithm contains a try-and-error process for cropping, in which the *nucleus periphery* labels are forced to appear with the *nucleus center* labels.

• Supplementary file 2. Comparison of linking performances Linking performances on the PH dataset, on a total number of 259,071 links (including 688 links on cell divisions). Incremental training was performed by transferring the training parameters from the model pre-trained with the CE datasets. Linking performance on dividing cells is scored separately.

• Supplementary file 3. Datasets used for training generic detection models Datasets used in training of the detection models used in *Figure 3*. Columns correspond to the datasets analysed in *Figure 3* and rows indicate the image datasets included in the training. In each case, the test image data were excluded from training. The Fluo-C3DH-A549 (*Castilla et al., 2019*), Fluo-C3DH-H157 (*Maška et al., 2013*), Fluo-N3DH-CHO (*Dzyubachyk et al., 2010*), Fluo-N3DH-CE (*Murray et al., 2008*) datasets are from the Cell Tracking Challenge (*Maška et al., 2014*).

• Supplementary file 4. Parameters used for training and prediction using StarDist3D Parameters used for training and prediction in the StarDist3D model used in *Figure 3—figure supplement 1*. These parameters were extracted from "config.json" and "thresholds.json" generated by the software.

• Supplementary file 5. Parameters used for training and validation of generic models Parameters used for training and validation of the generic models used in *Figure 3*. Size and scale are represented in the format [X]x[Y]x[Z].

• Supplementary file 6. Parameters used for fine-tuning and prediction using generic models Parameters used for fine-tuning of the generic models and prediction used in *Figure 3*. Size and scale are represented in the format [X]x[Y]x[Z].

• Transparent reporting form

## Data availability

The imaging datasets are available at https://doi.org/10.5281/zenodo.4630933. The source code for the ELEPHANT software is available at https://github.com/elephant-track.

The following dataset was generated:

| Author(s) | Year | Dataset title | Dataset URL | Database and Identifier |
|---|---|---|---|---|
| Sugawara, Çevrim and Averof | 2021 | PH_li13 Zenodo | http://doi.org/10.5281/zenodo.4630933 | Zenodo, 10.5281/zenodo.4630933 |

The following previously published datasets were used:

| Author(s) | Year | Dataset title | Dataset URL | Database and Identifier |
|---|---|---|---|---|
| Waterston Lab | 2008 | C. elegans developing embryo | http://data.celltrackingchallenge.net/training-datasets/Fluo-N3DH-CE.zip | Cell Tracking Challenge, Fluo-N3DH-CE |
| Kamm R | 2017 | MDA231 human breast carcinoma cells | http://data.celltrackingchallenge.net/training-datasets/Fluo-C3DL-MDA231.zip | Cell Tracking Challenge, Fluo-C3DL-MDA231 |

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
