## [Decision Letter]

**Decision letter after peer review:**

Thank you for submitting your article "Tracking cell lineages in 3D by incremental deep learning" for consideration by *eLife*. Your article has been reviewed by 3 peer reviewers, and the evaluation has been overseen by a Reviewing Editor and Aleksandra Walczak as the Senior Editor. The following individuals involved in review of your submission have agreed to reveal their identity: Asim Iqbal (Reviewer #1); Christian Tischer (Reviewer #3).

Essential revisions:

1) Please address the usability of the code (i.e., please see all three reviewers comments and try to address them).

2) Please address reviewer comments regarding if a 3DUNet is sufficient, and truly state of the art; please provide results comparing to other architectures (such as StarDist, as reviewer 2 points out, or other architectures, see reviewer 1). Also, please consider adding limitations around network options to the discussion.

3) Multiple reviewers have questions around the ellipse tracker – could you show this is best / alternatives for cells with other shapes, and please consider a limitation regarding this (i.e., I assume a neuron with a long axon would not be well fit by an ellipse).

*Reviewer #1 (Recommendations for the authors):*

– Consider moving figure 1 in supplementary and make figure 2 as figure 1.

– Add a block diagram to show how to use ELEPHANT step by step through an example.

– It would be nice to show training, validation curves of 3D U-Nets in the supplementary figures to confirm if there is no over-fitting in the models.

– Demonstrate the usage and performance of the framework on diverse examples in the main figures (e.g. Figure 2).

– Expand the Result section in the study.

*Reviewer #2 (Recommendations for the authors):*

More details about running the code: The specific link to the documentation was broken on github. I tried to go through the instructions in the readme, discovered that Mastodon installation requires java sdk (these instructions should be included for linux, windows and mac). Once java was installed the user needs to convert tiffs, like those from celltrackingchallenge.net, into bigdataviewer format. This procedure should be documented at least for the format provided on celltrackingchallenge. For instance, I opened a folder of tiffs in imageJ, converted to a stack, then saved as an xml. If there is a way to do this for instance from the command line it would be great for it to be documented. Next I opened the xml from inside mastodon (the window that pops up from "java -jar elephant-0.1.0-client.jar"). I was able to open bdv but when I tried to do anything in the elephant plugin it said connection refused. I was running the code on Ubuntu 18.04.

It would be helpful to include more discussion about the amount of data needed, and the amount of manual input. This tool has increased practical value if ~1 month of interactive tracking (as described in the paper) is not needed for each dataset. It is excellent that the networks in the paper are provided as pth files. Can you have the networks as options in the mastodon plugin so that users can easily access them?

Can you comment on the use of ellipses to approximate nuclei instead of more complex shapes? Is the advantage of this representation that it is easy to use in the case of sparse labels? Or do you see it as advantageous to allow overlapping masks? Similarly, for the optical flow model, the output of the detection model is used to compute optical flow, so ellipses instead of precise cell boundaries. Have you considered how having precise cell boundaries might help the optical flow model perform better?

*Reviewer #3 (Recommendations for the authors):*

Overall I think this is fantastic work and I would be very happy to review a revised version of the software.

30: I would just write "..for 3D cell tracking.."

37: "in a crustacean (1 week)" It is not clear to me what the "1 week" refers to. Maybe the number of time points and cells would be more informative in this technical context?

63: It is not really "based on Fiji", maybe write "deployed in Fiji"?

194: "To reduce the amount of…" Does one also need to duplicate the data when running client and server on the same computer? For big image data it would be very nice to avoid this.

214: Insert space between "without "

215: "showed non-negligible variations in intensity" Is this a problem for the deep learning detection model? If so, this should be elaborated on and a section "Image data preparation" where this is explained should be added to the documentation.

224: "On the server, images, annotation labels and outputs were stored in the Zarr format," I am curious: Why is it necessary to store the image in Zarr format rather than HDF5?

226: "these data were prepared using a custom Python script" Running a python script within a Docker container could be quite a hurdle for non-computational end-users. Any chance that could be simplified?

247: "*L*prioris" There is a space missing.

ELEPHANT/Mastodon software and documentation

Mastodon

The author's software should be compatible with the latest version of Mastodon, which includes a few bug fixes that avoid hanging of the software during the annotation process.

Example demo data set

To get started, the authors provide an example data set, which is great. However, for me training the detection and linkage on the current example data set takes too much time to be done during a review of the publication. I would appreciate if the authors provided a much simpler demo dataset where everything (detection + linkage) could be done within maximally 30 minutes of work. I think for reviewing the software and also for beginner users such a toy data set would be extremly useful.

Server connection

I think adding something to the user interface that makes the connection to the server more explicit would be very nice.

For example: Plugins > ELEPHANT > Connect to Server

Then one could put functionality there that would, e.g., allow the user to check whether the connection is working and maybe some feedback about which server one is connecting to.

In fact, for connecting to the Google Colab sever one should explore whether it is possible to create a UI in Mastodon where the user could just copy and paste these two lines:

SSH command: ssh -p10739 root@8.tcp.ngrok.io

Root password: qXzK8cOwvkWxdAcGZwM0

And then the Java code would parse those two lines create system calls to establish the server connection via the two SSH commands. This would be much more convenient than the current workflow where one needs to open a terminal and modify tedious SSH command line calls (also, many less IT savvy users could be put off by the command line calls).

Maybe for the other server modes similar ideas could be explored (personally I only looked into the Colab based solution).

It would be great if there was more feedback within the client on what is happening right now on the server side. I added specific suggestions in few places (see below). One could even consider mirroring all the text output that is generated server side in the Elephant client log window.

Training of detection

While I think I get the point now, it is a bit though to understand all the different tags (TP,FP,…).

What I understood now is that probably it is OK to simply add spots manually and they would be used as training data (being tagged as TP by default). If that is true I would suggest to split the annotation workflow in the documentation in a basic and advanced version, where in the basic version one maybe does not need to explicitly provide manual tags at all?!

https://elephant-track.github.io/#/v0.1/?id=_2-shortcuts

Current text: If you cannot find the ~/.mastodon/keymaps/ directory, please run [File > Preferences…] first to create it with the ~/.mastodon/keymaps/keymaps.yaml.

Suggested text: If you cannot find the ~/.mastodon/keymaps/ directory, please run [File > Preferences…] and click [OK] to create it. Please restart Mastodon for the Elephant keymap to become active.

In addition, it would really be great if setting up the keymap.yaml file was easier.

One could for example provide the already edited keymap.yaml file for download and tell the user to replace the current one. Since you are shipping a stand-alone version of Mastodon anyway, even better would be if that was somehow included in (or taken care of by) the elephant.jar. Could you somehow ship this information inside the jar?

https://elephant-track.github.io/#/v0.1/?id=detection-workflow

I would recommend adding a sentence here that first the connection to the server needs to be established.

https://elephant-track.github.io/#/v0.1/?id=_5-establish-connections-from-your-computer-to-the-server-on-colab

It would be nice to add an explanation why one needs to establish two connections (rather than only one).

https://elephant-track.github.io/#/v0.1/?id=_2-initialize-a-model

It would be very good if there was more feedback within the Mastodon UI about whether and when the model initialization has finished successfully.

Also feedback about the training progress, e.g. the decrease of the loss, the current cycle, a progress bar, would be great such that one can judge how well the training worked and whether the current number of training cycles is adequat.

Typo in Mastodon: "Detection > Reset *a* Seg Model". I suggest removing the "a".

"Predicted spots and manually added spots are tagged by default as unlabeled and fn, respectively."

I wonder whether manually added spots should be tagged as tp by default? At least I often forgot clicking "4" to mark them as tp. In fact, I am confused now, because maybe the manually added spots are tagged as tp by default?

https://elephant-track.github.io/#/v0.1/?id=_6-importing-and-extending-a-pretrained-model

Importing a pretrained model is simple. Just specify the model parameter file located at the workspace/models in the settings.

I could not figure out where to specify the model parameter file. On the client or on the server? And how to do it exactly?

[Editors' note: further revisions were suggested prior to acceptance, as described below.]

Thank you for resubmitting your work entitled "Tracking cell lineages in 3D by incremental deep learning" for further consideration by *eLife*. Your revised article has been reviewed by 3 peer reviewers and the evaluation has been overseen by Aleksandra Walczak as the Senior Editor, and a Reviewing Editor.

The manuscript has been improved but there are some remaining issues that need to be addressed, as outlined below:

Please be sure to not use the term "state of the art" (SOTA) unless you demonstrate truly best performance (which you do not) – it is not a requirement to be SOTA to be published. Moreover, please address reviewer #2's request, and consider reviewer #3, i.e., providing local GPU instructions (vs only COLAB).

*Reviewer #1 (Recommendations for the authors):*

Thanks to the authors for submitting the revised manuscript and providing the response to the reviewers' comments. The manuscript, as well as the codebase, are significantly updated after taking the feedback from the reviewers into account, in particular, figure 3 is a useful addition in the manuscript and it showcases the performance of ELEPHANT on diverse datasets. A systematic comparison between ELEPHANT and StarDist 3D is also useful to evaluate the performance comparison.

However, the limited performance of ELEPHANT on segmentation tasks is expected since the method is limited to detect ellipsoid shape-based objects but since the method is focused on only detection and tracking so it would be useful to state it clearly in the abstract and manuscript. This will help the users to get a better idea about the strengths and limitations of the toolbox in advance. Overall the study seems to be in much better shape now.

*Reviewer #2 (Recommendations for the authors):*

Thank you for the really great improvements to usability. I was able to easily install Elephant and Mastodon through Fiji. The google colab server setup took around 30 minutes to get started – I'm not sure if there's any way to make it faster, but wanted to point it out. After that I tried to "add port forward" and received a "Connection refused" error, there was no pop up to input my password. Is there another step with rabbitMQ permissions perhaps that I'm missing?

Thanks for also running StarDist on one of the frames. Can you please add quantitative metrics to Supplementary Figure 8? Maybe they are somewhere but I missed them and apologies if I did. Given StarDist does not have temporal information, it is likely that Elephant outperforms StarDist, but it would be good to include the quantitative results for the reader to be able to decide whether to use StarDist or Elephant. Thanks for the information about how stardist+trackmate are only in 2D.

*Reviewer #3 (Recommendations for the authors):*

First of all we would like to congratulate the authors for doing a great job in addressing the issues that we have raised in the previous review. As a result the software is in our view now much more user friendly; for example connecting from the Fiji user interface to the deep learning server is a great improvement as compared to the previous command line based way.

However, in practice we still struggled to reliably work with the Google Colab server and we feel that this might be a source of frustration for the potential users of the software. In the previous version of the software the authors also presented another solution (i.e. a local server), given that the users would have a computer with an appropriate GPU. Maybe one could reconsider those ideas?

We are also wondering, given the advances in running deep learning models in Java (DeepImageJ and CSDBDeep) whether a fully Java based (i.e. one Fiji plugin) solution would be feasible to make this great tool more user friendly and stable? We know that this would not solve the issue of providing the GPU resources, but maybe users would then simply need to have a computer with a GPU (which we think could be "fair enough").

---

## [Author Response]

Essential revisions:1) Please address the usability of the code (i.e., please see all three reviewers comments and try to address them).

We have implemented many changes that improved the usability of the code, following the reviewers' suggestions (see our responses to individual reviewers' comments, below).

2) Please address reviewer comments regarding if a 3DUNet is sufficient, and truly state of the art; please provide results comparing to other architectures (such as StarDist, as reviewer 2 points out, or other architectures, see reviewer 1). Also, please consider adding limitations around network options to the discussion.

ELEPHANT is state of the art because it makes deep learning available for cell tracking/lineaging in the absence of extensive annotated datasets for training. To achieve this, the network architecture must support training with sparse annotation. 3DUNet is adapted to this strategy. Existing deep learning applications which employ other architectures, such as StarDist, do not currently fulfil this purpose.

We now show that ELEPHANT models trained with sparse annotations perform similarly well to trained StarDist3D models for nuclear detection in single 3D stacks (see Figure 3—figure supplement 1). For cell tracking over time, StarDist and Trackmate have so far only been implemented in 2D and could therefore not be used on our 3D datasets.

As we describe in the paper, using the 3DUNet architecture, ELEPHANT outperformed a large number of other tracking applications in the Cell Tracking Challenge (http://celltrackingchallenge.net/latest-ctb-results/) in both cell detection and linking (tracking). This comparison includes methods that employ deep-learning.

ELEPHANT is less performant than other algorithms in segmentation, which is not the software's main purpose. We explain this point in the revised manuscript, page 7.

3) Multiple reviewers have questions around the ellipse tracker – could you show this is best / alternatives for cells with other shapes, and please consider a limitation regarding this (i.e., I assume a neuron with a long axon would not be well fit by an ellipse).

We use ellipsoids for annotation because they are essential for rapid and efficient training and predictions, which are the backbone of interactive deep learning (see detailed response to reviewer 2). This is in fact acknowledged as one of the strengths of our method by reviewer 3. We now explain this in page 4 of the revised manuscript.

We provide additional data showing that using ellipsoids is sufficient for detection of elongated and irregularly-shaped cells using ELEPHANT (Figure 3E). Post-processing can be appended to our workflow if a user needs to extract the precise morphology of the tracked nuclei or cells.

Reviewer #1 (Recommendations for the authors):– Consider moving figure 1 in supplementary and make figure 2 as figure 1.

Incremental training is the key concept of our method and we want to present this in figure 1.

– Add a block diagram to show how to use ELEPHANT step by step through an example.

We thank the reviewer for this suggestion. We have added a block diagram as Figure 1—figure supplement 2. We also provide a demo dataset and we are preparing a step by step video tutorial, which will be embedded in the user manual (also see response to reviewer 3).

– It would be nice to show training, validation curves of 3D U-Nets in the supplementary figures to confirm if there is no over-fitting in the models.

We have included training and validation curves for a pair of CE datasets, which confirm there is no significant over-fitting in the model (Figure 3—figure supplement 2). We present these results in the revised manuscript, page 5.

– Demonstrate the usage and performance of the framework on diverse examples in the main figures (e.g. Figure 2).– Expand the Result section in the study.

We have analysed three additional datasets with different characteristics, on which we provide the tracking performance of a pre-trained model as well as the dramatic improvement in performance obtained by the addition of sparse annotations. We present the results in the main text (page 5) and in new Figure 3.

The new datasets we used are:

– Intestinal organoids acquired by confocal microscopy (Kok et al. 2020 PLoS One doi:10.1101/2020.03.18.996421)

– Intestinal organoids acquired by light-sheet microscopy (de Madeiros et al. 2021 bioRxiv doi:10.1101/2021.05.12.443427)

– MDA231 human breast carcinoma cells, including elongated and irregularly-shaped cells (Fluo-C3DL-MDA231 dataset from Cell Tracking Challenge)

Reviewer #2 (Recommendations for the authors):More details about running the code: The specific link to the documentation was broken on github. I tried to go through the instructions in the readme, discovered that Mastodon installation requires java sdk (these instructions should be included for linux, windows and mac). Once java was installed the user needs to convert tiffs, like those from celltrackingchallenge.net, into bigdataviewer format. This procedure should be documented at least for the format provided on celltrackingchallenge. For instance, I opened a folder of tiffs in imageJ, converted to a stack, then saved as an xml. If there is a way to do this for instance from the command line it would be great for it to be documented. Next I opened the xml from inside mastodon (the window that pops up from "java -jar elephant-0.1.0-client.jar"). I was able to open bdv but when I tried to do anything in the elephant plugin it said connection refused. I was running the code on Ubuntu 18.04.

We thank the reviewer for raising these practical issues that matter to the users. We have addressed these issues as follows:

– The practical issue with the broken link has been fixed.

– ELEPHANT is now available as an extension on Fiji. The Fiji environment automatically installs all the components needed to run ELEPHANT/Mastodon, with no need for the users to install additional packages. We agree that this will greatly facilitate the use of ELEPHANT by non-expert users.

– We prepared a command line tool for converting Cell Tracking Challenge data into datasets usable on ELEPHANT/Mastodon. Fiji also provides a user-friendly GUI tool for this purpose.

– In addition, we have introduced numerous changes to facilitate user interactions, particularly through a control panel that allows users to set up the server and to monitor its status.

– We have added a demo dataset and pre-trained models that users can use to test ELEPHANT (explained in the updated user manual).

– We are in the process of producing a video tutorial, which will be embedded in the user manual and available on YouTube.

It would be helpful to include more discussion about the amount of data needed, and the amount of manual input. This tool has increased practical value if ~1 month of interactive tracking (as described in the paper) is not needed for each dataset. It is excellent that the networks in the paper are provided as pth files. Can you have the networks as options in the mastodon plugin so that users can easily access them?

We now show that a generic pre-trained model can be re-trained with a very modest amount of new annotations – less than 10 nuclei, requiring an annotation time of a few minutes – to achieve a very high level of precision and recall in most datasets (Figure 3). This is shown with diverse image datasets (*C. elegans* embryo, *Parhyale* regenerating limbs, human intestinal organoids, breast carcinoma cells, captured either by confocal or light sheet microscopy).

Within ELEPHANT, we now provide a pre-trained model that has been trained on a wide range of image datasets. Users can start their tracking using this pre-trained model, which provides a basic level of performance.

All trained models are saved and can be re-used for subsequent tracking by selecting the required model in the Preferences panel (the generic pre-trained model is used as a default). Users can also share and access trained models through URL links (explained in the updated user manual).

Can you comment on the use of ellipses to approximate nuclei instead of more complex shapes? Is the advantage of this representation that it is easy to use in the case of sparse labels? Or do you see it as advantageous to allow overlapping masks? Similarly, for the optical flow model, the output of the detection model is used to compute optical flow, so ellipses instead of precise cell boundaries. Have you considered how having precise cell boundaries might help the optical flow model perform better?

As we now explain in the manuscript (page 4), we use ellipsoids for annotation because they are essential for rapid and efficient training and predictions, compared with complex shapes, which require a larger number of parameters to describe them. This is essential for interactive (real-time) training. In practice, using ellipsoids also reduces the amount of work required for annotating the data, compared with precise drawing of cell outlines. Ellipsoids also take less memory space than more complex annotations in the tracked data files.

Ellipsoids indeed allow overlapping masks, which could be an advantage when dealing with complex objects under low spatial resolution.

To clarify the process followed in ELEPHANT: the optical flow maps are generated directly from the images, with no input from the detected ellipsoids; the annotated nuclei are linked subsequently by associating the flow maps with the detected nuclei (see Methods).

ELEPHANT has currently been built as an extension of Mastodon, which uses ellipsoids for cell annotation. In this framework we are not able to test and compare the potential impact of using precise cell annotations, but this might be relevant to test in a different framework.

Reviewer #3 (Recommendations for the authors):Overall I think this is fantastic work and I would be very happy to review a revised version of the software.30: I would just write "..for 3D cell tracking.."

Corrected.

37: "in a crustacean (1 week)" It is not clear to me what the "1 week" refers to. Maybe the number of time points and cells would be more informative in this technical context?

We have added the number of timepoints.

63: It is not really "based on Fiji", maybe write "deployed in Fiji"?

Corrected.

194: "To reduce the amount of…" Does one also need to duplicate the data when running client and server on the same computer? For big image data it would be very nice to avoid this.

Duplicating the data is currently required because Mastodon only supports the HDF5/XML format while the ELEPHANT server supports the Zarr format. This point will be addressed by future updates of the software.

214: Insert space between "without "

Corrected.

215: "showed non-negligible variations in intensity" Is this a problem for the deep learning detection model? If so, this should be elaborated on and a section "Image data preparation" where this is explained should be added to the documentation.

On the server, the image data are stored in unsigned 8-bit or unsigned 16-bit format, keeping the original image format. At the beginning of processing on the server, the image data are automatically converted to a 32-bit float and their intensity is normalized at each timepoint in a way that the minimum and maximum values become 0 and 1. The text in this part of the manuscript refers to normalization on the client, which is helpful for a better visualization on Mastodon. This is now explained in the revised manuscript in the section 'Dataset preparation'.

224: "On the server, images, annotation labels and outputs were stored in the Zarr format," I am curious: Why is it necessary to store the image in Zarr format rather than HDF5?

We chose this format because we expected that Zarr would be more rapid than HDF5 (see Moore et al. 2021, bioRxiv 2021.03.31.437929; Kang et al. 2019, Proc Int Conf High Performance Computing, Networking, Storage and Analysis), which is preferable when the ELEPHANT server is deployed on the cloud environment. However we have not compared the speed of these formats directly.

226: "these data were prepared using a custom Python script" Running a python script within a Docker container could be quite a hurdle for non-computational end-users. Any chance that could be simplified?

In the latest version of ELEPHANT, this conversion can also be done from the client application. This is explained in the revised manuscript, page 11.

247: "*L*prioris" There is a space missing.

Corrected.

ELEPHANT/Mastodon software and documentationMastodonThe author's software should be compatible with the latest version of Mastodon, which includes a few bug fixes that avoid hanging of the softare during the annotation process.

After introducing some updates on both Mastodon and ELEPHANT, the software is now implemented on the latest version of Mastodon. ELEPHANT is also now deployed in Fiji.

Example demo data setTo get started, the authors provide an example data set, which is great. However, for me training the detection and linkage on the current example data set takes too much time to be done during a review of the publication. I would appreciate if the authors provided a much simpler demo dataset where everything (detection + linkage) could be done within maximally 30 minutes of work. I think for reviewing the software and also for beginner users such a toy data set would be extremly useful.

We now provide a new demo dataset that is a small subset of the CE1 dataset (see revised user manual).

Server connectionI think adding something to the user interface that makes the connection to the server more explicit would be very nice.For example: Plugins > ELEPHANT > Connect to ServerThen one could put functionality there that would, e.g., allow the user to check whether the connection is working and maybe some feedback about which server one is connecting to.

To facilitate the setting up of the ELEPHANT server, we have now implemented a control panel that allows users to monitor the process and provides links to the relevant section of the user manual and to Google Colab (see revised user manual).

In fact, for connecting to the Google Colab sever one should explore whether it is possible to create a UI in Mastodon where the user could just copy and paste these two lines:SSH command: ssh -p10739 root@8.tcp.ngrok.ioRoot password: qXzK8cOwvkWxdAcGZwM0And then the Java code would parse those two lines create system calls to establish the server connection via the two SSH commands. This would be much more convenient than the current workflow where one needs to open a terminal and modify tedious SSH command line calls (also, many less IT savy users could be put off by the command line calls).Maybe for the other server modes similar ideas could be explored (personally I only looked into the Colab based solution).

We have now implemented an SSH client that runs on ELEPHANT using the JSch library. Users can establish a connection to the server via SSH from the ELEPHANT Control Panel.

It would be great if there was more feedback within the client on what is happening right now on the server side. I added specific suggestions in few places (see below). One could even consider mirroring all the text output that is generated server side in the Elephant client log window.

In addition to the client log window, we now implement a server log window that duplicates the server-side text output to the client side.

Training of detectionWhile I think I get the point now, it is a bit though to understand all the different tags (TP,FP,…).What I understood now is that probably it is OK to simply add spots manually and they would be used as training data (being tagged as TP by default). If that is true I would suggest to split the annotation workflow in the documentation in a basic and advanced version, where in the basic version one maybe does not need to explicitly provide manual tags at all?!

We recognize that the colour tags would be overly complicated for most users, but we think that they can be useful for advanced users. As the reviewer suggests, we have therefore established two colour modes, which users can select depending on their needs (see user manual): a basic colour mode (which is the default) and an advanced colour mode. In the basic mode, only three colour tags are used ('accepted' in cyan, 'rejected' in magenta, 'predicted' in green). The original colors are kept for advanced usage (e.g. to visually inspect prediction results using True/False information). In both modes, False/True information is used in the deep learning to facilitate the training process by giving a higher weight to False annotations.

https://elephant-track.github.io/#/v0.1/?id=_2-shortcutsCurrent text: If you cannot find the ~/.mastodon/keymaps/ directory, please run [File > Preferences…] first to create it with the ~/.mastodon/keymaps/keymaps.yaml.Suggested text: If you cannot find the ~/.mastodon/keymaps/ directory, please run [File > Preferences…] and click [OK] to create it. Please restart Mastodon for the Elephant keymap to become active.In addition, it would really be great if setting up the keymap.yaml file was easier.One could for example provide the already edited keymap.yaml file for download and tell the user to replace the current one. Since you are shipping a stand-alone version of Mastodon anyway, even better would be if that was somehow included in (or taken care of by) the elephant.jar. Could you somehow ship this information inside the jar?

In the latest version of ELEPHANT, the default keymap settings are installed automatically. Users can switch the keymap settings via Mastodon’s Preferences dialog. The user manual has been updated accordingly.

https://elephant-track.github.io/#/v0.1/?id=detection-workflowI would recommend adding a sentence here that first the connection to the server needs to be established.

When the connection to the server fails, the latest version of ELEPHANT shows an error dialog with the message "The ELEPHANT server is unavailable. Please set it up first.". The availability of the server can be checked in the Control Panel.

https://elephant-track.github.io/#/v0.1/?id=_5-establish-connections-from-your-computer-to-the-server-on-colabIt would be nice to add an explanation why one needs to establish two connections (rather than only one).

The ELEPHANT server provides main functionalities (e.g. detection, linking), while the RabbitMQ server is used to send messages to the client (e.g. progress, completion). The updated version of the user manual explains this.

https://elephant-track.github.io/#/v0.1/?id=_2-initialize-a-modelIt would be very good if there was more feedback within the Mastodon UI about whether and when the model initialization has finished successfully.

The updated version of ELEPHANT shows the progress messages in the server log window.

Also feedback about the training progress, e.g. the decrease of the loss, the current cycle, a progress bar, would be great such that one can judge how well the training worked and whether the current number of training cycles is adequat.

Information on training progress is now available in the server log window. Advanced users can now get access to loss information through TensorBoard (instructions on how to do this will be given in a future update of the user manual).

Typo in Mastodon: "Detection > Reset *a* Seg Model". I suggest removing the "a".

The menu titles have been updated.

"Predicted spots and manually added spots are tagged by default as unlabeled and fn, respectively."I wonder whether manually added spots should be tagged as tp by default? At least I often forgot clicking "4" to mark them as tp. In fact, I am confused now, because maybe the manually added spots are tagged as tp by default?

Following the reviewer's suggestion, in the basic color scheme tp and fn are now both highlighted with the same colour (cyan).

Nuclei that are missed by prediction but subsequently annotated by the user are tagged as fn, which will turn into tp if they are predicted correctly in the next iteration. The distinction between tp and fn is still useful when examining the performance of the detection model. Users can access this information in the advanced mode.

https://elephant-track.github.io/#/v0.1/?id=_6-importing-and-extending-a-pretrained-modelImporting a pretrained model is simple. Just specify the model parameter file located at the workspace/models in the settings.I could not figure out where to specify the model parameter file. On the client or on the server? And how to do it exactly?

The model parameter file can be specified in the Preferences dialog on the client, where the file path is relative to “/workspace/models/” on the server. In the updated version of ELEPHANT, there are two ways to import the pre-trained model parameters:

1. Upload the pre-trained parameters file to the website that provides a public download URL (e.g. GitHub, Google Drive, Dropbox). Run [Plugins > ELEPHANT > Detection > Reset Detection Model] and select the “From URL” option with the download URL.

2. Directly place/replace the file at the specified file path on the server.

The updated version of the user manual includes this explanation.

[Editors' note: further revisions were suggested prior to acceptance, as described below.]

The manuscript has been improved but there are some remaining issues that need to be addressed, as outlined below:Please be sure to not use the term "state of the art" (SOTA) unless you demonstrate truly best performance (which you do not) – it is not a requirement to be SOTA to be published. Moreover, please address reviewer #2's request, and consider reviewer #3, i.e., providing local GPU instructions (vs only COLAB).

We do not apply this term to ELEPHANT, as requested. We apply the term only to benchmarked, best-performing software that we compared with ELEPHANT in the Cell Tracking Challenge.

We have addressed the relevant comments of reviewers #2 and #3, as requested.

Reviewer #1 (Recommendations for the authors):Thanks to the authors for submitting the revised manuscript and providing the response to the reviewers' comments. The manuscript, as well as the codebase, are significantly updated after taking the feedback from the reviewers into account, in particular, figure 3 is a useful addition in the manuscript and it showcases the performance of ELEPHANT on diverse datasets. A systematic comparison between ELEPHANT and StarDist 3D is also useful to evaluate the performance comparison.However, the limited performance of ELEPHANT on segmentation tasks is expected since the method is limited to detect ellipsoid shape-based objects but since the method is focused on only detection and tracking so it would be useful to state it clearly in the abstract and manuscript. This will help the users to get a better idea about the strengths and limitations of the toolbox in advance. Overall the study seems to be in much better shape now.

The title and abstract of the paper state clearly that ELEPHANT's objective is to enable or to facilitate cell tracking. We make no claims about cell or nuclei segmentation.

Reviewer #2 (Recommendations for the authors):Thank you for the really great improvements to usability. I was able to easily install Elephant and Mastodon through Fiji. The google colab server setup took around 30 minutes to get started – I'm not sure if there's any way to make it faster, but wanted to point it out. After that I tried to "add port forward" and received a "Connection refused" error, there was no pop up to input my password. Is there another step with rabbitMQ permissions perhaps that I'm missing?

We have included a video tutorial in the user manual, which shows how to set up ELEPHANT with the Google Collab option. We also provide some online support and advice on troubleshooting to users through the image.sc forum.

We have reduced the time required for setup to around 15 minutes. The Google Collab option is sometimes restricted/failing due to GPU quota and connection timeout. We now provide tips on how to resolve these issues in the user manual.

Thanks for also running StarDist on one of the frames. Can you please add quantitative metrics to Supplementary Figure 8? Maybe they are somewhere but I missed them and apologies if I did. Given StarDist does not have temporal information, it is likely that Elephant outperforms StarDist, but it would be good to include the quantitative results for the reader to be able to decide whether to use StarDist or Elephant. Thanks for the information about how stardist+trackmate are only in 2D.

We provide the relevant metrics in Figure 3—figure supplement 1.

Reviewer #3 (Recommendations for the authors):First of all we would like to congratulate the authors for doing a great job in addressing the issues that we have raised in the previous review. As a result the software is in our view now much more user friendly; for example connecting from the Fiji user interface to the deep learning server is a great improvement as compared to the previous command line based way.However, in practice we still struggled to reliably work with the Google Colab server and we feel that this might be a source of frustration for the potential users of the software. In the previous version of the software the authors also presented another solution (i.e. a local server), given that the users would have a computer with an appropriate GPU. Maybe one could reconsider those ideas?

We still provide the option with a local server. The instruction can be found in the user manual at the section 'Advanced options for the ELEPHANT Server' (https://elephant-track.github.io/#/v0.3/?id=advanced-options-for-the-elephant-server)

The corresponding system requirements can be also found in the user manual (https://elephant-track.github.io/#/v0.3/?id=system-requirements).

We have kept the following description in the manuscript: "The server environment is provided as a Docker container to ensure easy and reproducible deployment (https://github.com/elephant-track/elephant-server). The server can also be set up with Google Colab in case the user does not have access to a computer that satisfies the system requirements."

We are also wondering, given the advances in running deep learning models in Java (DeepImageJ and CSDBDeep) whether a fully Java based (i.e. one Fiji plugin) solution would be feasible to make this great tool more user friendly and stable? We know that this would not solve the issue of providing the GPU resources, but maybe users would then simply need to have a computer with a GPU (which we think could be "fair enough").

We agree that it would be interesting to explore a fully Java based solution. This would require extensive re-writing of the software. We are considering this for future updates.